# 9-O-Terpenyl-Substituted Berberrubine Derivatives Suppress Tumor Migration and Increase Anti-Human Non-Small-Cell Lung Cancer Activity

**DOI:** 10.3390/ijms22189864

**Published:** 2021-09-13

**Authors:** Jia-Ming Chang, Jin-Yi Wu, Shu-Hsin Chen, Wen-Ying Chao, Hsiang-Hao Chuang, Kam-Hong Kam, Pei-Wen Zhao, Yi-Zhen Li, Yu-Pei Yen, Ying-Ray Lee

**Affiliations:** 1Department of Surgery, Division of Thoracic Surgery, Ditmanson Medical Foundation Chiayi Christian Hospital, Chiayi City 60002, Taiwan; 06016@cych.org.tw (J.-M.C.); 07137@cych.org.tw (K.-H.K.); 2Department of Physical Therapy, College of Medical and Health Science, Asia University, Taichung 41354, Taiwan; 3Department of Medical Research, Ditmanson Medical Foundation Chiayi Christian Hospital, Chiayi City 60002, Taiwan; 10472@cych.org.tw (S.-H.C.); 14421@cych.org.tw (P.-W.Z.); 10862@cych.org.tw (Y.-Z.L.); 15159@cych.org.tw (Y.-P.Y.); 4Department of Microbiology, Immunology and Biopharmaceuticals, College of Life Sciences, National Chiayi University, Chiayi 60004, Taiwan; jywu@mail.ncyu.edu.tw; 5Department of Nursing, Min-Hwei College of Health Care Management, Tainan 73658, Taiwan; A129@o365.mhchcm.edu.tw; 6Division of Pulmonary and Critical Care Medicine, Department of Internal Medicine, Kaohsiung Medical University Hospital, Kaohsiung Medical University, Kaohsiung 80708, Taiwan; hsianghao.chuang@gmail.com; 7Department of Microbiology and Immunology, College of Medicine, Kaohsiung Medical University, Kaohsiung 80708, Taiwan

**Keywords:** berberrubine derivatives, non-small cell lung cancer, anticancer, synthesis, autophagic flux, tumor migration

## Abstract

Lung cancer is one of the most common cancers and the leading cause of death in humans worldwide. Non-small-cell lung cancer (NSCLC) accounts for approximately 85% of lung cancer cases and is often diagnosed at a late stage. Among patients with NSCLC, 50% die within 1 year after diagnosis. Even with clinical intervention, the 5-year survival rate is only approximately 20%. Therefore, the development of an advanced therapeutic strategy or novel agent is urgently required for treating NSCLC. Berberine exerts therapeutic activity toward NSCLC; therefore, its activity as an antitumor agent needs to be explored further. In this study, three terpenylated-bromide derivatives of berberrubine were synthesized and their anti-NSCLC activities were evaluated. Each derivative had higher anti-NSCLCs activity than berberrubine and berberine. Among them, 9-O-gernylberberrubine bromide (B4) and 9-O-farnesylberberrubine bromide (B5) showed greater growth inhibition, cell-cycle regulation, in vitro tumorigenesis suppression, and tumor migration reduction. In addition, some degree of apoptosis and autophagic flux blocking was noted in the cells under B4 and B5 treatments. Our study demonstrates that the berberrubine derivatives, B4 and B5, exhibit impressive anti-NSCLC activities and have potential for use as chemotherapeutic agents against NSCLC.

## 1. Introduction

Lung cancer is one of the most common and lethal types of cancers in both men and women, accounting for approximately 2 million new cases and 1.8 million cancer-related deaths per year [1]. Patients are diagnosed with lung cancer usually at a late stage. Among these patients, >50% die within 1 year [1]. The main subtype of lung cancer is non-small-cell lung cancer (NSCLC), which accounts for approximately 85% of lung cancer cases [1]. Currently, the treatments available for NSCLC include physical surgery, chemotherapy, radiation therapy, targeted therapy, immunotherapy, and a combination of these therapies. The 5-year survival rate for patients who receive clinical intervention is only approximately 17.8% [1]. Although target therapy and immunotherapy are prospective approach in the NSCLC patients, occurrence of off-target or resistance of target therapy, and treatment-related deaths and/or discontinuations because of pneumonitis, colitis in immunotherapy are still the challenges [2,3]. Therefore, continual investigation and development of advanced therapeutic strategies and novel therapeutic compounds are urgently needed to improve the overall survival rate in patients with NSCLC.

Berberine and berberrubine are isoquinoline alkaloids derived from the plants *Berberis aristata* and *B.*
*vulgari*s L, respectively; berberrubine is readily derived from berberine [4]. Berberine and berberrubine possess broad-spectrum pharmacological properties. They have been used for the treatment of microbial infection, diabetes, hyperlipidemia, diarrhea, and cardiovascular and psychotic diseases as well as anti-inflammatory and antioxidative agents [5]. These compounds exhibit a selective cytotoxicity against various human cancer cells, including hepatoma, colorectal carcinoma, breast cancer, melanoma, and NSCLC [6,7,8].

The anti-NSCLC activities of berberine and berberrubine include proliferation inhibition, tumorigenesis suppression, cell-cycle regulation, and cell death induction (in vitro and in vivo) [8,9]. In addition, berberine inhibits the migration and invasion of tumor cells [10,11]. However, pharmacokinetic studies have indicated that berberine has poor oral bioavailability and increasing the dose often elicits gastrointestinal side effects [12]. Because of its hydrophilic nature, berberine is absorbed poorly in the intestines and thus exhibits a low inhibitory activity in the suppression of cancer cell growth [13]. To improve its intestinal absorption and anticancer activity, we synthesized a series of berberine derivatives and identified their cytotoxicities in hepatoma, colon, and bladder cancer cells [7,14]. We have previously demonstrated that a series of 9-O-alkylberberrubine derivatives possess higher anti-NSCLC activities than berberine and berberrubine [8]. In the present study, we synthesized a series of 9-O-terpenylberberrubine derivatives and evaluated their antitumor activities in NSCLC cell lines and compared them with those of berberine and berberrubine. All derivatives showed greater NSCLC cell growth inhibition than the parent compound. We further investigated the underlying mechanisms. Tumor metastasis in NSCLC poses a *challenge* in its treatment and influences disease prognosis. We determined the efficacy of the derivatives in inhibiting tumor migration in NSCLC cells. This study developed novel berberrubine-derived agents that exhibited satisfactory anti-NSCLC activities in eliminating tumor growth and metastasis.

## 2. Results

### 2.1. NSCLC Cell Growth Inhibition Induced by Berberrubine and Its Derivatives

Berberine (B1); berberrubine (B2); and several 9-terpenylated-bromide derivatives of berberrubine, including 9-O-isoprenylberberrubine bromide (B3), 9-O-gernylberberrubine bromide (B4), and 9-O-farnesylberberrubine bromide (B5) (Figure 1), were synthesized by Dr. Jin-Yi Wu. The growth inhibition of human NSCLC cell lines A549, H23, and H1435 induced by these compounds was examined. All compounds exhibited partial to significant growth inhibition of NSCLC cells in a dosage- and time-dependent manner (Figure 2). Berberine (B1) and berberrubine (B2) weekly inhibited the growth of NSCLC cells (Figure 2). In contrast, 9-O-gernylberberrubine bromide (B4) and 9-O-farnesylberberrubine bromide (B5) strongly inhibited the growth of NSCLC cells (Figure 2 and Table 1). The half-maximal inhibitory concentration (IC_50_) values (μM) of B4 and B5 did not vary significantly, suggesting that B4 and B5 had similar toxicity toward NSCLC cells (Table 1). Sensitivity toward B4 or B5 treatment was noted to be the highest in A549 cells, followed by that in H1435 cells and then in H23 cells (Figure 2 and Table 1). Although the treatments of B1 and B3 in A549 cells, and B2 and B3 in H1435 cells exhibited slightly lower cell viability at 24 h post-treatment comparing with 48 h post-treatment (Figure 2), there were no significant difference between the growth curve.

### 2.2. B5-Induced In Vitro Tumorigenesis Suppression in Human NSCLC Cells

Although B4 and B5 exhibited similar growth inhibition abilities in the three NSCLC cell lines, B5 seem more toxicity than B4 in H23 cells (Table 1). We further evaluated the effect of B5 on the tumorigenesis of NSCLC cells using an in vitro colony formation assay. The results demonstrated that administration of B5 significantly suppressed tumorigenesis in A549, H23, and H1435 cells and that this phenomenon was associated with the incubation concentration (Figure 3). Interestingly, B5-induced inhibition of tumorigenesis was higher in A549 and H23 cells than that in H1435 cells (Figure 3). This finding shows that B5 possesses satisfactory anti-NSCLC properties that are valuable in the suppression of cancer tumorigenesis.

### 2.3. B5-Induced Cell-Cycle Regulation in Human NSCLC Cells

Cell-cycle regulation and cell death are responsible for growth inhibition in anticancer therapy. We investigated cell-cycle regulation of NSCLC cells under B5 treatment. A flow cytometry analysis was performed for cells incubated with B5; the data revealed that B5 treatment induced cell-cycle arrest at the G0/G1 phase in all NSCLC cells in a dose-dependent manner (Figure 4). We further verified the expression of cell-cycle regulation proteins, including phospho-CDK2, CDK4, and p27, in cells incubated with B5. The data showed a decreased expression of phospho-CDK2 and CDK4 and an increased expression of p27 in all three cell lines under B5 treatment (Figure 4). These results demonstrate that B5 administration can induce G0/G1 arrest in NSCLC cells.

### 2.4. Partial Apoptosis Induction in B5-Treated Human NSCLC Cells

To address whether B5 regulates cellular apoptosis in NSCLC cells, the cells were treated with B5 and then cellular phenomena were examined using optical microscopy. Taxol was used as a positive control and an inducer of apoptosis. No significantly apoptotic A549 and H1435 cells were noted after B5 treatment (Figure 5A,C). However, partially apoptotic cells were detected in the H23 cell line after B5 treatment (Figure 5B). To investigate the activation of caspase-3 and PARP in cells under B5 treatment, western blot analysis was performed. The data demonstrated that B5 induced the partial activation of PARP in H23 cells; however, this phenomenon was not noted in A549 or H1435 cells (Figure 5B). No significant caspase-3 activation was observed in any of the cells under B5 treatment (Figure 5). We suggest that B5 administration can partially induce cellular apoptosis in H23 cells but not in A549 and H1435 cells; this activity of B5 may be mediated through a caspase-independent pathway.

### 2.5. Autophagy Modulation in B5-Treated NSCLC Cells

Autophagy is an intracellular self-degradative mechanism that can maintain homeostasis and intracellular energy by disassembling unnecessary or dysfunctional cellular components. It plays a dual role in cancers by promoting tumor growth and survival through the maintenance of cellular energy and by eliminating stress. It has been recognized as an alternative therapeutic strategy in cancer treatment [15]. Therefore, we investigated whether B5 could regulate autophagy in human NSCLC cells. A549, H23, and H1435 cells were incubated with B5, and western blot analysis was performed to examine the autophagic biomarker, LC3. LC3-II was upregulated in these cells depending on B5 concentration after its administration (Figure 6A), which suggests that B5 can modulate cellular autophagy. Because autophagy is generally maintained in basal cellular physiology, the activation of autophagy increases autophagosome formation, followed by further fusion with lysosomes to form autolysosomes, which digest proteins, lipids, and the organelles in the their cargos. Increasing the level of LC3-II may induce autophagy or block autophagic flux. Our study confirmed the expressions of LC3-II and p62, on a time axis, in B5-treated cells. Only H23 cells exhibited a simultaneous increase and decrease of LC3-II and p62 levels (Figure 6B), which suggests that B5 can induce autophagy activation in H23 cells but block autophagic flux in A549 and H1435 cells. Further confirmation was sought by transfecting the cells with a plasmid, pmRFP-EGFP-LC3, to monitor the process of autophagic flux. The data revealed that rapamycin improved the expression of autophagosome and autolysosome puncta in A549 cells. However, chloroquine inhibited the fusion of autophagosomes with lysosomes, resulting in the accumulation of former (Figure 6C). B5-treated A549 cells exhibited autophagosome accumulation (Figure 6C). Most puncta were autolysosomal in the cells treated with rapamycin in combination with B5 (Figure 6C), which demonstrated that B5 suppressed autophagic flux in A549 cells under rapamycin treatment. To address the biological effect of B5-mediated autophagic flux inhibition in A549 cells, autophagy was enhanced or inhibited by a combination of rapamycin, 3-MA or chloroquine with B5 and cellular phenomena and viability were examined. Autophagy induction in A549 cells treated with rapamycin and B5 reduced the total cell number (Figure 7). Interestingly, in A549 cells, treatment with the combination of B5 and rapamycin showed more toxicity than that by B5 or rapamycin alone (Figure 7). Moreover, blocking endogenous autophagy with 3-MA suppressed cellular viability; however, a combination of B5 and 3-MA significantly reversed B5-mediated cellular toxicity (Figure 7). In addition, blocking endogenous autophagic flux with chloroquine reduced the total cell number, and this effect as the autophagic flux inhibitor, B5, in A549 cells. Therefore, combination treamtnet with chloroquine and B5 could enhance cellualr toxicity in A549 cells. These findings demonstrate that B5-blocked autophagic flux in A549 cells has an anticancer function and enhancing autophagy or autophagic flux in A549 cells can increase the anticancer activity of B5. However, suppressing initiated cellualr autophagy with 3-MA in A549 cells can reduce B5 mediated anti-cancer activity.

### 2.6. B5-Induced Tumor Migration Suppression in Human NSCLC Cells

Tumor metastasis in NSCLC poses a challenge for disease treatment and influences its prognosis. Reducing or blocking the migration of NSCLC is a therapeutic strategy to improve anticancer effect and patient survival. Therefore, we evaluated the effect of B5 on tumor migration in NSCLC using a wound-healing migration assay and a trans-well migration assay. Treatment with B5 reduced cellular migration in A549 cells (Figure 8A,B). Moreover, using TGF-β to induce a mesenchymal phenotype in A549 cells (Figure 8C) enhanced migration of A549 cells (Figure 8B); B5 suppressed this TGF-β-induced tumor migration (Figure 8B). In addition, it reduced mesenchymal markers (N-cadherin and snail) and increased epithelial markers (E-cadherin and claudin-1) in A549 cells (Figure 8C), suggesting that B5 can reduce tumor migration through modulated mesenchymal–epithelial transition in NSCLC.

## 3. Discussion

NSCLC is an aggressive, malignant tumor associated with high global mortality [1]. Surgery and chemotherapy are the first-line treatments. However, patients commonly present with recurrence and metastasis; the 5-year survival is <20% [1]. Therefore, a highly efficacious therapeutic agent or strategy is urgently needed for NSCLC treatment. Previous studies have revealed that berberine—a compound obtained from the roots, bark, stems, and rhizomes of *Berberis* sp.—is used in Chinese traditional medicine. At high doses, it can effectively combat human malignant cancers, including NSCLC [16]. Due to the hydrophilic nature of berberine, it is difficult to pass cell membrane and increasing the dose often elicits side effects, resulting poor bioavailability [12]. Therefore, modification of berberine to improve its bioavailability and enhance its toxicity toward human cancer cells is an interesting strategy. We modified and synthesized a series of 9-*O*-terpenyl-substituted derivatives of berberrubine and demonstrated that 9-*O*-gernylberberrubine bromide (B4) and 9-*O*-farnesylberberrubine bromide (B5) markedly inhibited the growth of NSCLC cells including cell cycle arrest and partially-induced apoptosis. Moreover, B5 blocked autophagic flux in A549 and H1435 cells, and modulation of endogenous autophagy under B5 treatment contributing to cellular toxicity in NSCLC cells. Importantly, B5 dramatically suppressed NSCLC cell migration by modulating the epithelial–mesenchymal transition.

Genetic alteration is an important strategy for malignant tumors. *p53* is the most frequent target for NSCLC, and mutations of *p53* appear to contribute toward the poor prognosis in NSCLC [17,18]. *P53* plays an important role in the cellular stress response pathways that respond to DNA damage and repair and in cell-cycle regulation [17,18]. In this study, the cell line A549 represented the *p53* wild type. The cells lines H23 (M246I) and H1435 (C141W) exhibit mutant forms of *p53* [19,20]. Growth inhibition in A549 and H1435 cells was more sensitive to B5 treatment (Table 1) than that in H23 cells; however, A549 and H23 cells were more sensitive to tumorigenesis suppression under B5 treatment (Figure 3). This implies that B5 treatment not only affects cell proliferation but also perturbs colony activity. In addition, KRAS mutation is also frequency found in lung adenocarcinoma, blocking of autophagy under suppressing of KRAS activity can elevate cytotoxicity of lung cancer cells [21]. Moreover, modulation of Keap1 and PTEN can regulate cellular autophagy in lung carcinoma cells [22]. Importantly, KRAS mutation are found in A549 and H23 cells; Keap1 mutation also can be found in A549 and H1435 cells, and PTEN mutation are exhibited in H23 cells. These findings suggest that the antitumor activities of B5 (growth inhibition and tumorigenesis suppression) in various NSCLC cell lines arise through divergent mechanisms; the diverse genetic backgrounds of the cells perhaps account for the results.

Pent et al. showed that berberine, at a high concentration (>40 μM), inhibits the growth of A549 cells; at a low dose (2.5 μM), it suppresses tumor migration and invasion [16]. Regarding this, our findings were consistent with those of the study by Pent et al. (Figure 2). Notably, the synthesized compounds B3, B4, and B5 exhibited higher toxicity than berberine (B1) or berberrubine (B2) toward NSCLC cells. Inhibition of tumor migration occurred at a lower concentration of B5 (1 μM) than that used in the abovementioned study (2.5 μM) [16].

The cell cycle controls cell division and proliferation and is regulated by several cyclin-dependent kinases. Regulation of cell-cycle checkpoints—by inducing cell-cycle arrest and inhibiting cell proliferation—is considered a therapeutic target in cancer treatment [23]. Berberine leads to cell-cycle arrest at the G1 phase in the treatment of various cancers [24,25]. Consistent with these findings, our results show that B5 also induces cell-cycle arrest at the G1 phase in the three NSCLC cell lines used in our study. The fundamental bioactivities of berberine and B5 are similar, but B5 exerts enhanced anticancer effects.

Berberine has also been reported to be a potent chemosensitizer and chemoprotector for conventional cancer therapies; it sensitizes tumor cells to chemotherapy and radiotherapy [26]. Berberine modulates various signaling pathways, including PI3K/Akt, MAPKs, AMPK, STAT3, NF-κB, EGFR, ROS, and HIF-1α, to sensitize cancer cells to a broad spectrum of cancer therapies [26]. In addition, blocking of programmed cell death-1 (PD-1)/programmed cell death ligand-1 (PD-L1) has become a major cancer immunotherapy in various human cancers including NSCLCs [27]. Interestingly, administration with berberine can improve anticancer immunity through suppressing the deubiquitination activity of constitutive photomorphogenesis 9 (COP9) signalosome subunit 5 (CSN5), resulting in ubiquitination and degradation of PD-L1 [27]. Whether B4 or B5 can also regulate these signaling pathways to improve sensitization to chemotherapy, radiotherapy and immunotherapy is an interesting issue that requires further investigation.

Autophagy is a cellular homeostasis program that can degrade of harmful or surplus cellular contents such as aggregated protein, dysfunctional/long-lived organelles, intracellular pathogens, and storage nutrients [28]. Several studies have described autophagy as a double-edged sword that acts to stimulate as well as suppress tumor growth [15,29,30,31]. Autophagy has been shown to be associated with drug resistance in various cancers [32]. Berberine can reverse doxorubicin resistance in breast cancer cells by regulating the PI3K/Akt pathway, and it can further attenuate autophagy by reducing the number of autophagosomes, enhancing the expression of p62, and inhibiting the expression of multidrug resistance protein 1 [33]. We demonstrated that B5 can modulate cellular autophagy in various cell lines (Figure 6 and Figure 7). Notably, elevating endogenous autophagy with rapamycin or enhancing autophagic flux block with chloroquine can increase B5-mediated cellular toxicity (Figure 7), suggesting that modulation of endogenous autophagy could improve anti-NSCLCs activity of B5. However, the underlying mechanisms involving B5 and the autophagy signaling pathways warrant further studies in the future.

Berberine not only contributes to cancer therapy but also has reported benefits in the treatment of inflammation, diabetes, and pathogenic infection and can attenuate acute renal injury and neurodegenerative diseases [34,35,36,37,38,39]. We synthesized B4 and B5 from berberrubine, derived from berberine; both B4 and B5 exert higher anti-NSCLC activities than berberine and berberrubine. Whether B4 or B5 can function against the diseases and dysfunctions discussed in this article should be an interesting topic for future exploration.

## 4. Materials and Methods

### 4.1. Chemical and Reagents

Berberine (B1), berberrubine (B2), 9-O-isoprenylberberrubine bromide (B3), 9-O-gernylberberrubine bromide (B4), and 9-O-farnesylberberrubine bromide (B5) were synthesized and provided by Dr. Jin-Yi Wu [7]. Dimethyl sulfoxide (DMSO), propidium iodide, crystal violet, chloroquine, rapamycin, *3-methyladenine* (3-MA), paclitaxel (Taxol), and transforming growth factor beta (TGF-β) were purchased from Sigma-Aldrich (St. Louis, MO, USA). Primary antibodies such as phospho-CDK2 (2351-1; Epitomics; Burlingame, CA, USA), CDK4 (sc-601; Santa Cruz Biotechnology; Dallas, TX, USA), p27 (GTX100446; GeneTex; Irvine, CA, USA), caspase-3 (#9662; Cell Signaling Technology; Beverly, MA, USA), PARP (#9542; Cell Signaling Technology), GAPDH (GTX100118; GeneTex), LC3 (AP1802a; Abgent; San Diego, CA, USA), p62 (#5114; Cell Signaling Technology), N-cadherin (#4061; Cell Signaling Technology), E-cadherin (610182; BD Biosciences; Bedford, MA, USA), claudin-1 (#4933; Cell Signaling Technology), and snail (#3879; Cell Signaling Technology) as well as secondary antibodies such as rabbit antimouse antibody (GTX26728; GeneTex) and goat antirabbit antibodies (GTX213110; GeneTex) were purchased and used in this study. All chemicals and biochemicals used in this study were of analytical grade.

### 4.2. Cell Lines and Cell Cultures

Human lung adenocarcinoma cell lines—A549, H23, and H1435—were purchased from the American Type Culture Collection (Manassas, VA, USA) and cultured, respectively, in Dulbecco’s Modified Eagle Medium (DMEM), RPMI medium, and DMEM mixed (1:1) with F-12K medium containing 10% fetal bovine serum (FBS), l-glutamine (2 mM), streptomycin (100 μg/mL), and penicillin (100 IU/mL; all from Gibco-Invitrogen, Carlsbad, CA, USA). Cells were incubated at 37°C in a humidified atmosphere containing 5% CO_2_.

### 4.3. Cellular Viability Assay

Cells were seeded into 96-well plates (5 × 10^3^ cells/well) and grown in the aforementioned media overnight. After attachment, cells were incubated with a control medium containing 0.01% DMSO or a medium with specific agents. After treatment, cellular viability was determined using the cell counting kit 8 (CCK-8) assay (Enzo Life Sciences, Farmingdale, NY, USA). Three independent assays were performed.

### 4.4. Western Blot Analysis

After cells were incubated with the control medium or that containing specific agents, the cell lysates were collected after lysing with M-PERTM protein extraction reagent (Thermo Fisher Scientific, Rockford, IL, USA) containing a 0.1% protease inhibitor cocktail. The sample proteins were loaded and then separated by sodium dodecyl sulfate–polyacrylamide gel electrophoresis and transferred to polyvinylidene fluoride membranes. Protein expressions were examined using the primary and secondary antibodies. The detailed experimental procedure has been described in our previous study [31].

### 4.5. Cell Cycle Analysis

Cells were cultured in the control medium or that with specific agents. The cells were stained using propidium iodide-containing RNase for 30 min in dark conditions at room temperature. Cell-cycle distribution was assessed using the FACScan^™^ flow cytometer (Becton Dickinson, San Diego, CA, USA). DNA content was analyzed using the Modfit LT 3.3 software (Verity Software House, Topsham, ME, USA). The detailed experimental procedure has been described in our previous study [40].

### 4.6. Colony Formation Assay

Cells were seeded in 12-well plates (1 × 10^3^ cells/well) and incubated overnight at 37°C. The cells were then incubated with the control medium or that with specific agents. After 12 days of incubation, the colony-forming efficiency of the cells and colony morphologies were determined after staining with 10% crystal violet solution (Sigma-Aldrich). The numbers of colonies were counted.

### 4.7. Autophagic Flux Observation

To observe the formation of autophagosome and autolysosome in cells treated with specific agents, a pmRFP-EGFP-LC3 plasmid (Addgene, Watertown, MA, USA) [31] was transfected into the cells using lipofectamine 2000 (Thermo Fisher Scientific) according to the manufacturer’s instructions. After the cells were incubated with the control medium or that with specific compounds, the autophagosome (yellow) and autolysosome (red) puncta in the cells were observed using confocal scanning microscopy (LSM 800; ZEISS, Oberkochen, Germany).

### 4.8. Migration Assay

To determine the impact of the novel compounds on NSCLC cell migration, a wound-healing migration assay was performed using a trans-well migration model. Wound-healing assay provided an easy and simple means for monitoring directional cell migration and interaction in vitro. The detailed procedure for the assay has been described in our previous publication [41]. The wound areas were detected microscopically. In the trans-well migration assay, the cells (1 × 10^5^ cells/well) were seeded and cultured in the upper chamber of the trans-well for 24 h; FBS was used as a chemoattractant in the bottom chamber for 24 h. The migrated cells in the bottom chamber were stained with 2% crystal violet solution. The detail procedure for this assay has been described in our previous study [42]. The migrated cells were observed under optical microscopy at a magnification of 100×.

### 4.9. Statistical Analysis

Data are presented as means ± standard errors for the indicated number of separate experiments. Statistical differences were determined using one-way ANOVA and Fisher’s least significant difference tests. Statistical significance was set at *p* < 0.05.

## Figures and Tables

**Figure 1 ijms-22-09864-f001:**
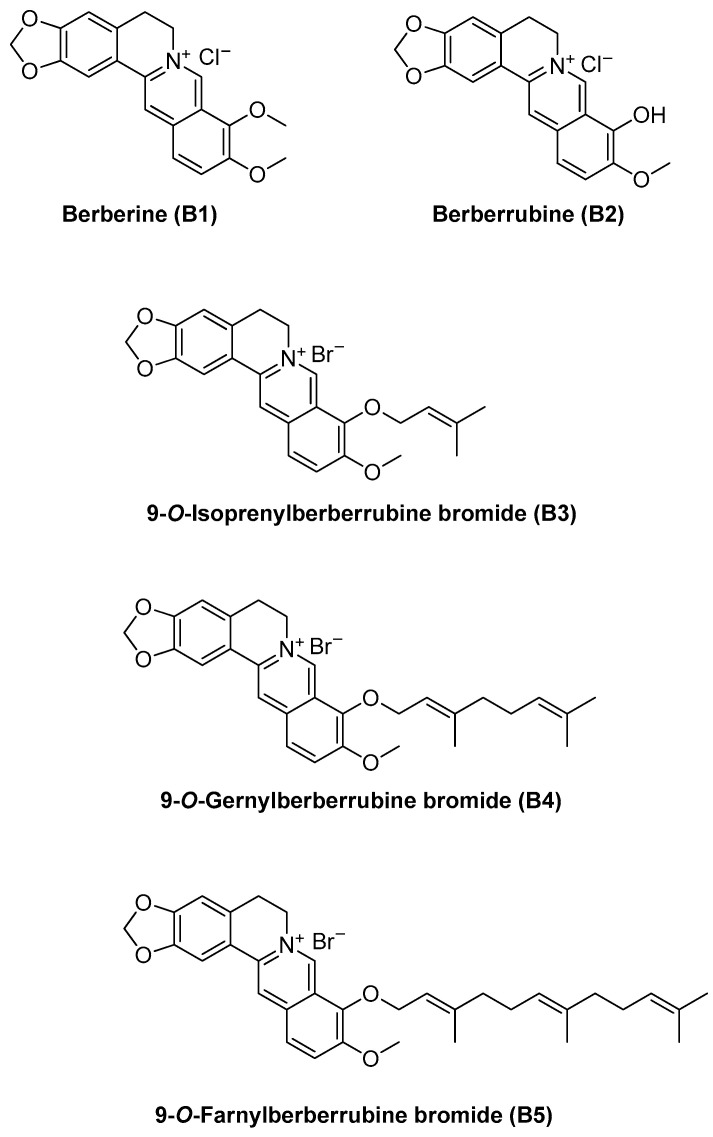
Chemical structures of berberine as well as berberrubine and derivatives.

**Figure 2 ijms-22-09864-f002:**
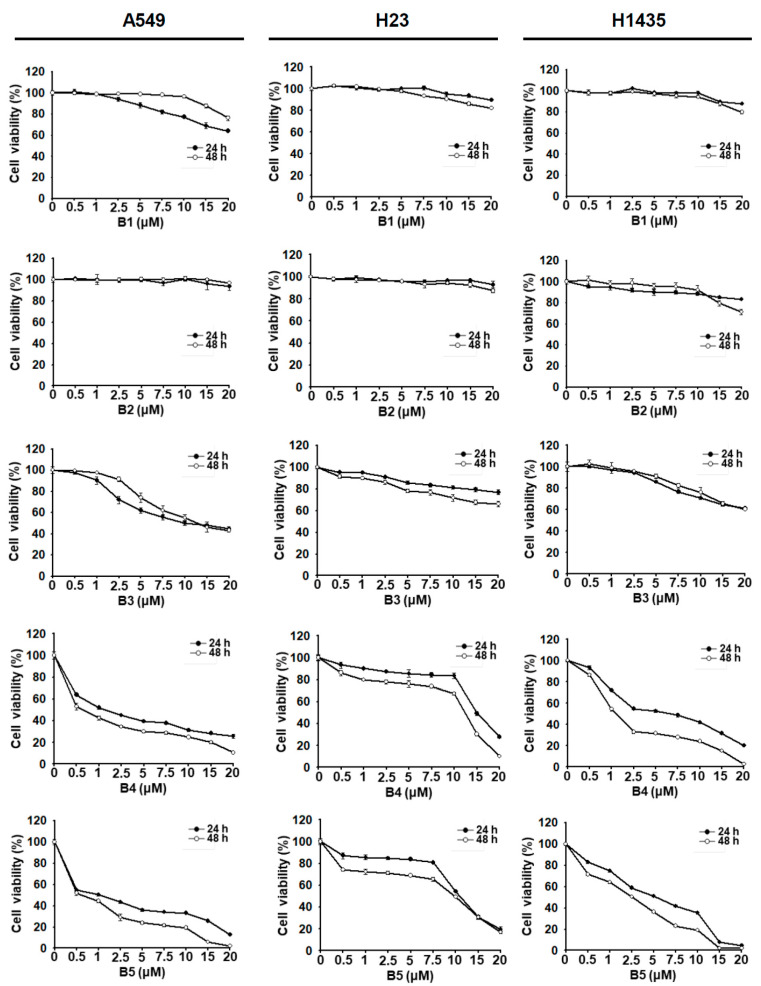
Berberine as well as berberrubine and its derivatives regulated the survival of NSCLC cells. A549, H23, and H1435 cells were treated with a control medium or a medium containing specific agents in various concentrations. Cell viability was tested using the CCK-8 assay at 24 h and 48 h after incubation.

**Figure 3 ijms-22-09864-f003:**
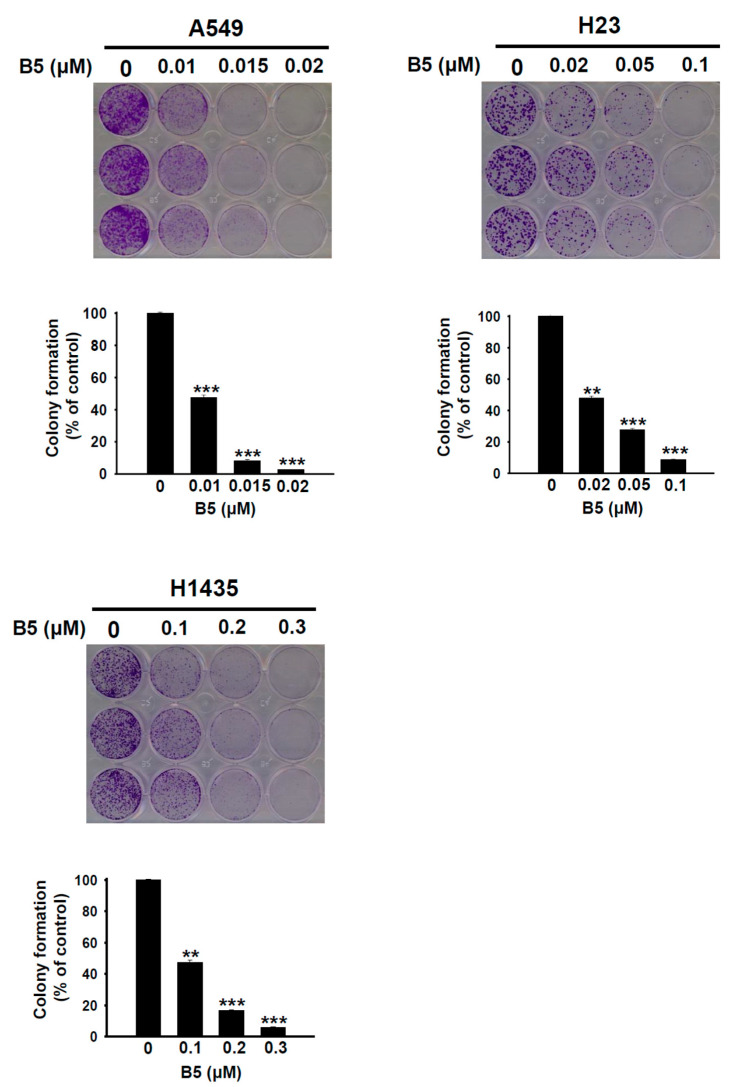
9-*O*-farnesylberberrubine bromide suppresses tumorigenesis in NSCLC cells in vitro. A549, H23, and H1435 cells were incubated with a control medium or that containing B5. Tumorigenesis was examined in vitro using a colony formation assay; the colonies were counted after staining. ** indicates *p* < 0.01; *** indicates *p* < 0.001.

**Figure 4 ijms-22-09864-f004:**
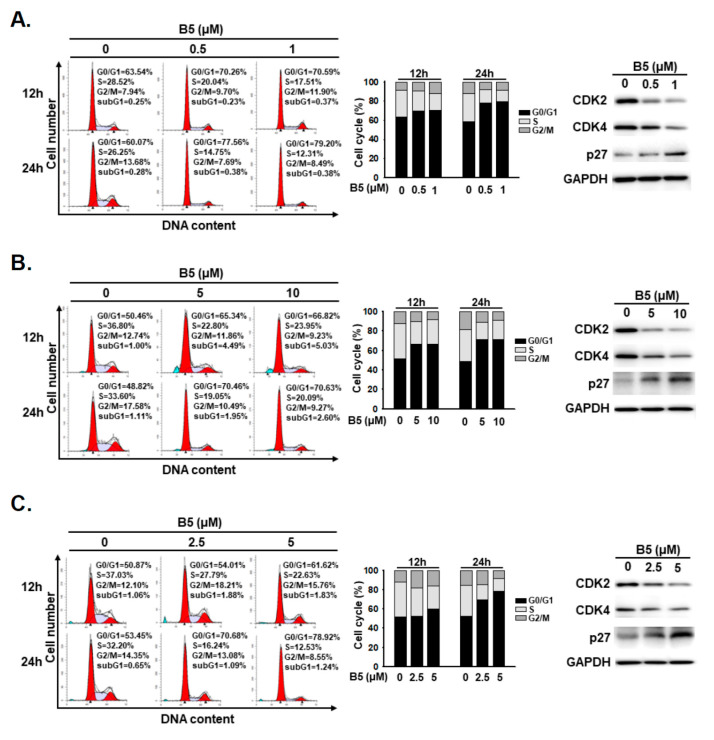
9-*O*-farnesylberberrubine bromide induces cell-cycle arrest in NSCLC cells. (**A**) A549, (**B**) H23, and (**C**) H1435 cells were incubated with a control medium or that containing B5 for 12 h and 24 h. The cell cycle was studied using flow cytometry. The expression of CDK2, CDK4, and p27 in the total cell lysate after the 24-h treatment was examined using western blot analysis. GAPDH was used as a loading control. Three independent experiments were performed, and a representative experiment was shown.

**Figure 5 ijms-22-09864-f005:**
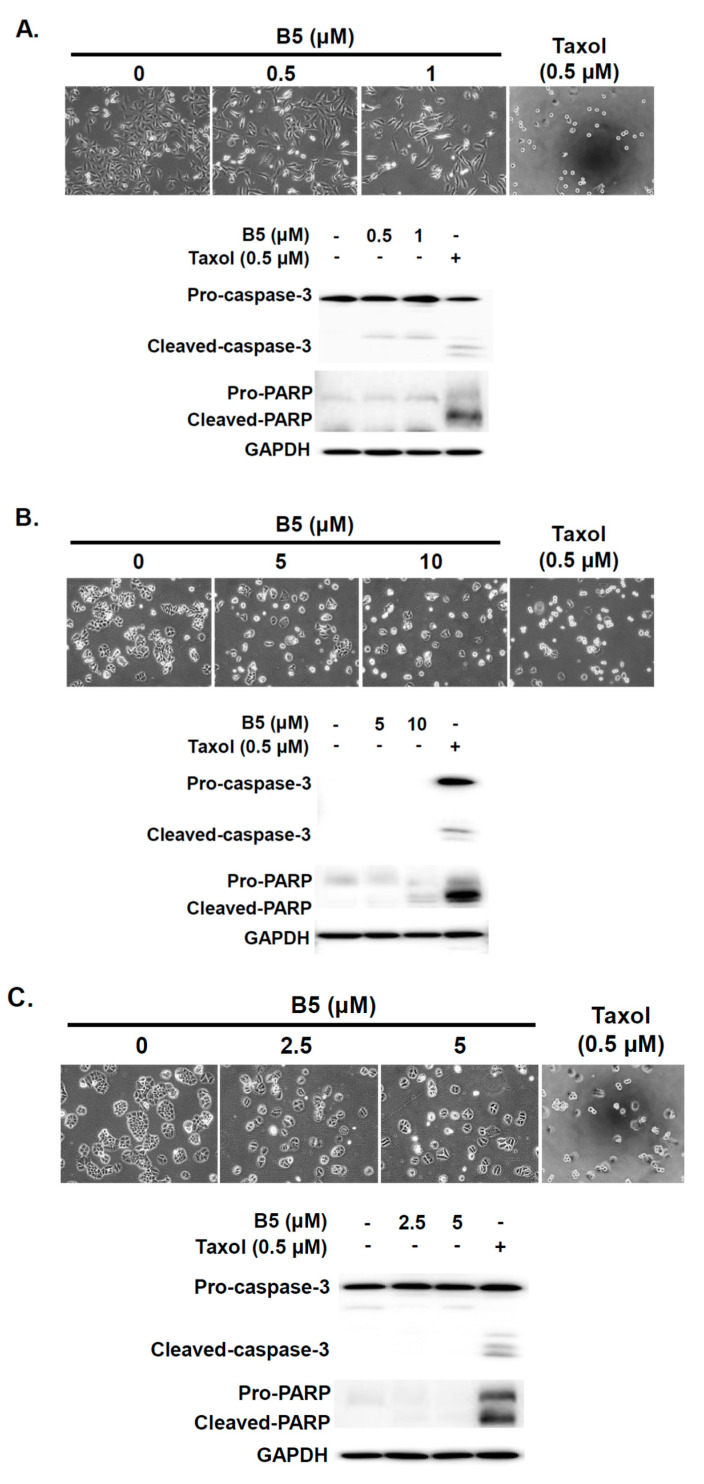
9-O-farnesylberberrubine bromide modulates apoptosis in NSCLC cells. (**A**) A549, (**B**) H23, and (**C**) H1435 cells were incubated with a control medium or that containing B5; the cellular morphology was observed using optical microscopy. The total cell lysate was collected after 24 h of treatment, and the expression of caspase-3 and PARP was examined using western blot analysis. Taxol was used as a positive control. GAPDH was used as a loading control. Three independent experiments were performed, and a representative experiment was shown.

**Figure 6 ijms-22-09864-f006:**
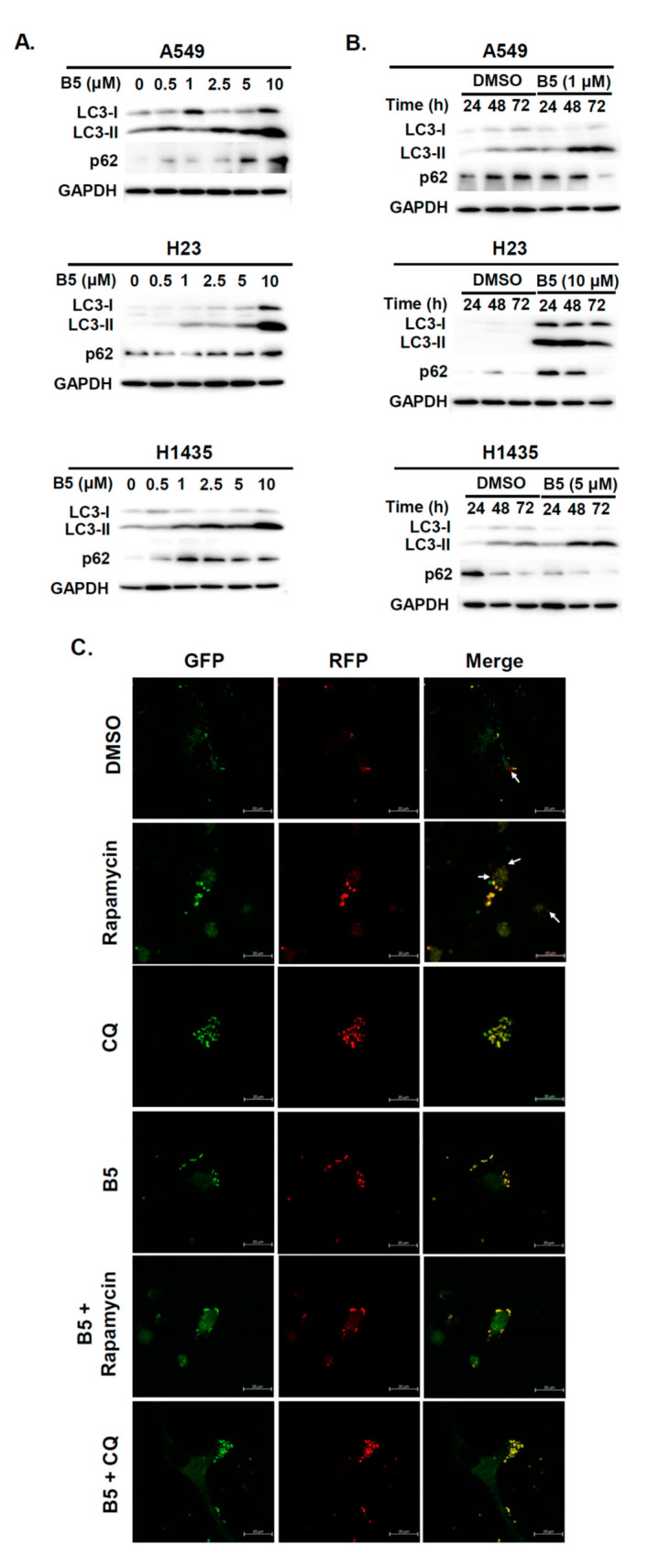
9-O-farnesylberberrubine bromide regulates autophagy in NSCLC cells. A549, H23, and H1435 cells were incubated a control medium or that containing B5 (**A**) in various doses and (**B**) at multiple times. The expression of LC3-I/LC3-II and p62 was detected using western blot analysis. GAPDH was used as a loading control. Three independent experiments were performed, and a representative experiment was shown. (**C**) Autophagic flux in A549 cells was examined after transfecting the cells with pmRFP-EGFP-LC3; the cells were then incubated with B5 (1 µM) or chloroquine (10 µM) for 48 h. Autophagosome (yellow) and autolysosome (red) puncta were observed using confocal microscopy. DMSO was used as a negative control. Rapamycin was used as an inducer of autophagy; chloroquine was a blocker of the autophagic flux.

**Figure 7 ijms-22-09864-f007:**
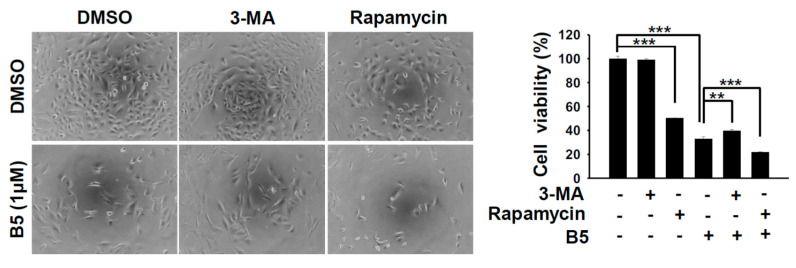
Enhancing autophagy further sensitizes A549 cells toward 9-O-farnesylberberrubine bromide treatment. A549 cells were incubated with B5, and autophagy was modulated with rapamycin, 3-MA or chloroquine. Cellular morphology and viability were examined using optical microscopy and CCK-8 assay in the cells administered with B5 (1 µM), 3-MA (5 mM), chloroquine (10 µM), and rapamycin (30 µM) or a combination after 48 h of treatment. 3-MA was an initiated autophagy inhibitor. Chloroquine was an autophagic flux blocker. Rapamycin was used as an autophagy inducer. ** indicates *p* < 0.01; *** indicates *p* < 0.001.

**Figure 8 ijms-22-09864-f008:**
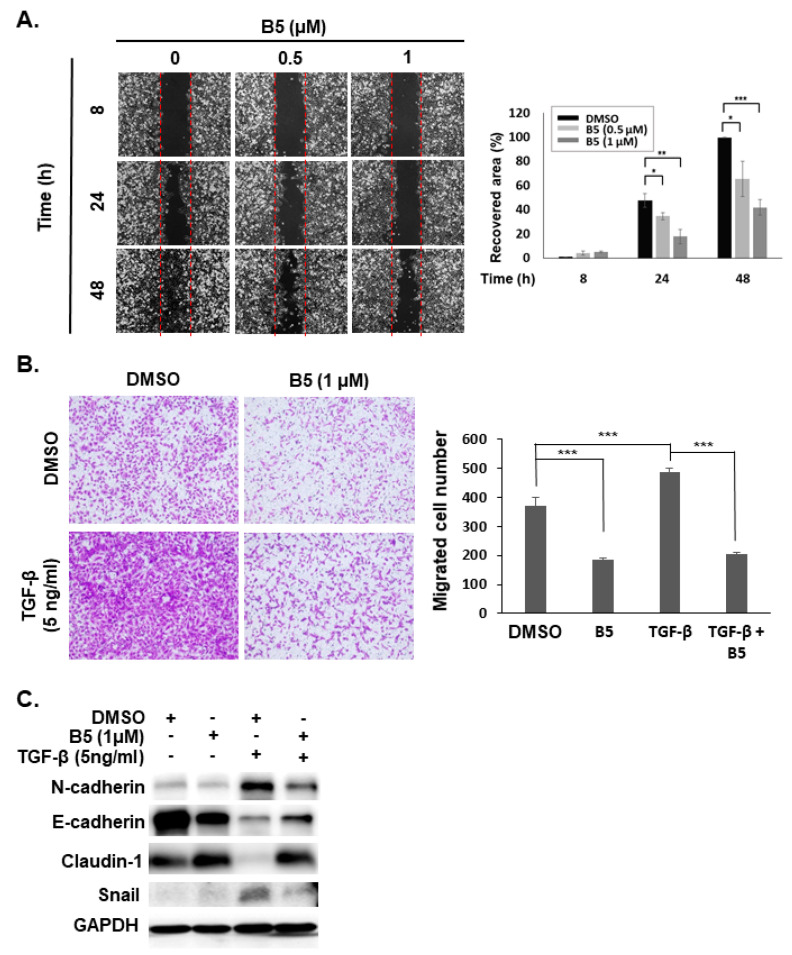
9-O-farnesylberberrubine bromide suppresses tumor migration in NSCLC cells. A549 cells were incubated with a control medium or that containing B5. (**A**) Wound-healing assay and (**B**) trans-well migration assay were performed to determine cellular migration using optical microscopy. *** indicates *p* < 0.001. (**C**) The expression of N-cadherin, E-cadherin, claudin-1, and snail in A549 cells was examined using western blot analysis after 48 h of treatment. DMSO was used as a negative control. TGF-β was used to induce the mesenchymal phenotype in A549 cells and improve A549 migration. GAPDH was used as a loading control. Three independent experiments were performed, and a representative experiment was shown. * indicates *p* < 0.05; ** indicates *p* < 0.01; *** indicates *p* < 0.001.

**Table 1 ijms-22-09864-t001:** The IC_50_ of berberine, berberrubine and its derivatives in human non-small-cell lung cancer cells.

Cell	Time (h)	IC_50_ (μM)				
		B1	B2	B3	B4	B5
A549	24 h	ND	ND	10.0 ± 2.1	1.4 ± 0.7	1.0 ± 0.4
	48 h	ND	ND	12.9 ± 2.8	0.6 ± 1.0	0.6 ± 0.2
H23	24 h	ND	ND	ND	14.9 ± 1.1	10.9 ± 1.7
	48 h	ND	ND	ND	12.4 ± 0.8	9.9 ± 0.1
H1435	24 h	ND	ND	ND	6.7 ± 0.6	5.1 ± 1.0
	48 h	ND	ND	ND	1.4 ± 0.5	2.6 ± 0.2

ND: not detected.

## Data Availability

All data used in this study are already provided in the manuscript at the required section. There are no underlying data available.

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
