# Peer review of "9-O-Terpenyl-Substituted Berberrubine Derivatives Suppress Tumor Migration and Increase Anti-Human Non-Small-Cell Lung Cancer Activity"

_ijms, 2021, doi:10.3390/ijms22189864_

Round 1

Reviewer 1 Report

The paper by Chang JM is an interesting work presented in a clear and concise manner, including even experiments with negative results, as the ones described in section 2.4. 

They evaluate the antitumor effect of three derivatives of berberrubine, a compound used in Traditional Chinese Medicine, which has the limitation of poor oral bioavailability. 

I consider that the topic is relevant and the article is of good quality, although there are some limitations to take into account:

  1. All the experiments are in vitro. Some in vivo experiments would greatly improve the work. The main limitation for the use of berberrubine was the poor oral bioavailability, it would be interesting to know if the berberrubine derivatives do not present this problem.
  2. It is surprising the lack of antitumor effect of B1 and B2 in figure 2. Did the authors expect this finding? Is there any explanation for the efficacy of the derivatives?
  3. Why were the experiments from section 2.2 performed only with B5 and not B4? They seem to have comparable antitumor effect in figure 2.
  4. The treatment with B5 seems to induce cancer cell differentiation towards a epithelial phenotype (decreased autophagy and mesenchymal markers, increased epithelial markers, reduced migration). It has been described that cancer cells can dedifferentiate to cancer stem cells (CSC) as a mechanism of resistance to PI3K/AKT/mTOR inhibition. This could explain the synergy of Rapamycin and B5, as the latter might be blocking the dedifferentiation. Could the authors explore these aspects evaluating some CSC markers in the treated cell lines? CD44/EpCAM, mitochondrial respiration and oxygen consumption, etc

Author Response

Reviewer 1

Q1. All the experiments are in vitro. Some in vivo experiments would greatly improve the work. The main limitation for the use of berberrubine was the poor oral bioavailability, it would be interesting to know if the berberrubine derivatives do not present this problem.

Response:

This is a very important comment. To address this issue, we are trying to study the pharmacokinetics of these compounds in vivo. We believe that our findings in this article will provide a new perspective of these compounds in biomedical investigation, not only in anticancers but also in microbial infection, diabetes, hyperlipidemia, diarrhea, cardiovascular and psychotic diseases, anti-inflammation and anti-oxidation.

Q2. It is surprising the lack of antitumor effect of B1 and B2 in figure 2. Did the authors expect this finding? Is there any explanation for the efficacy of the derivatives?

Response:

B1 is berberine and has been reported to exhibit anti-NSCLC activity (Toxicol Appl Pharmacol. 2006; 214:8-15.). B2 is berberrubine and has also been demonstrated to exert an anticancer activity in human colorectal carcinoma and hepatoma (Mol Pharmacol. 2002; 61:879-84; Fitoterapia. 2014; 92:230-7.). In the present study we evaluated the ani-NSCLCs activity of theses compounds, and B5 showed a most cytotoxicity effect in Figure 1 and Table 1. Therefore, we focuse to investigate the anti-NSCLC activity of B5 in the further study. Nevertheless, it remains interesting to investigate the biological effects of B1 to B4 in future studies.

Q3. Why were the experiments from section 2.2 performed only with B5 and not B4? They seem to have comparable antitumor effect in figure 2.

Response:

Berberine and berberrubine have anticancer activity is well known. Due to the hydrophilic nature of berberine, it is difficult to pass cell membrane and increasing the dose often elicits side effects, resulting poor bioavailability. In this dtudy, we modified berberrubine with 9-O-terpenyl to improve the cytotoxicity, and we evaluated the nati-NSCLC activity of these compounds. Although B4 and B5 exert similar anti-NSCLCs activity, B5 shows more toxicity in H23 cells than B4 (Table 1). Therefore, we decide to investigate the detail anti-NSCLC behavior of B5 in the further study. This instruction can be found in the lines  70-83, and 113-114.

Q4. The treatment with B5 seems to induce cancer cell differentiation towards a epithelial phenotype (decreased autophagy and mesenchymal markers, increased epithelial markers, reduced migration). It has been described that cancer cells can dedifferentiate to cancer stem cells (CSC) as a mechanism of resistance to PI3K/AKT/mTOR inhibition. This could explain the synergy of Rapamycin and B5, as the latter might be blocking the dedifferentiation. Could the authors explore these aspects evaluating some CSC markers in the treated cell lines? CD44/EpCAM, mitochondrial respiration and oxygen consumption, etc

Response:

This is a very interesting comment and opinion. Berberine has reported to promote the differentiation of hippocampal precursor cells and neurons (Animal Cells and Systems. 2008;12:203-209.), and in K562 cells (Zhong Yao Cai. 2009; 32:384-8). Therefore, we can’t rult out this possibility that the synergestic effect of rapamycin and B5 is through blocking of A549 cells dedifferentiation. If it is correct, it will provide a new vision and therapeutic strategy of B5 in anticancers. However, it needs further investigation. Importantly, in this study, rapamycin and 3-MA were used to address whether modulation of endogenous autophagy can affect anti-NSCLC activity in A549 cells under B5 treatment.

Reviewer 2 Report

In this work, the Authors present preliminary data on the role of berberrubine derivatives as potential antitumor agents in NSCLC. 

The work is original and interesting, although the results are maybe limited and won't have great scientific soundness.

I would suggest the Authors to provide a more complete and structured background on available treatment options for NSCLC, which at the present time include definitely more therapeutic strategies beyond surgery and chemotherapy. Starting from targeted therapy, which should be cited to be more comprehensive, I would dedicate a section on immunotherapy also because the Authors state that berberrubine derivatives have an anti-inflammatory role. In this context, it may be worthy to evaluate the role of these novel compounds in association with IO agents.

Reviewer 3 Report

The authors of the manuscript entitled “Synthesis of 9-O-terpenyl-substituted berberrubine derivatives suppress tumor migration and increase anti-human non-small-cell lung cancer activity” reported the synthesized berberrubine derivatives, mainly the 9-O-Farnesylberberrubine bromide (B5), inhibited cell growth, tumorigenesis and cell cycle progression of A549, H23 and H1435 cell lines. The inhibitory effects of the B5 compound may be through the induction of autophagy and blockage of autophagic flux. The authors also showed that the treatment of B5 compound also inhibited the in vitro migration of A549 cells. This is an extensive study of the work published in the Int J Mol Sci. 2020 Jun 13;21(12):4218 by the same research team. From these two studies, the 9-O-Decylberberrubine Bromide, 9-O-Dodecylberberrubine Bromide (B6 and B7 respectively in previous study) and 9-O-Farnesylberberrubine bromide (B5 in current study) showed very similar activity in A549 and H1435 cell lines. Although the author provided some evidences showing the negative effect of B5 on cell migration of A549, there is no significantly new finding in this study. Other

  1. In Figure 2, the treatment of B1 and B3 resulted in a lower cell viability of A549 for 24 h compared to the effect of same treatment on cell viability of A549 for 48 h. The authors should explain this phenomenon.
  2. In Table 1, please provide the standard deviation for each mean.
  3. From the results of cell viability assay (Figure 2 and Table 1), the B4 and B5 compounds had similar anticancer activity. However, all of the following experiments were done by using the B5 compound only. According to the title of this manuscript, the authors should also examine the effect of the B4 compound in the regulation of cell cycle, tumorigenesis, apoptosis, autophagy and migration.
  4. In Figure 7, 3-MA did not effectively reverse B5-inhibited cell growth, although it is statistically significant. The authors may try another autophagy inhibitors, such as chloroquine used in figure 6C, because 3-MA is also served as a PI3K inhibitor.
  5. For the in vitro migration assay, the quantification of the wound healing assay is required. In addition, 1 uM is the IC50 of B5 compound for 24 h treatment. It cannot be ruled out that the 1 uM of B5 treatment inhibits cell growth but not cell migration in the transwell migration assay. The authors should try lower concentration of B5 compound for these experiments.
  6. For the Discussion section, the authors should discuss the issue raised from the results of this study in more detail. Several suggestions are listed as following.

(1) In the first paragraph, the effect of B4 compound was not fully tested in this study, especially for the cell cycle arrest and apoptosis. The statement in line243-250 is not appropriate. What is the role of autophagy in the therapeutic effect of the B5 compound?

(2) For the second paragraph, in addition to the p53 mutations, the three cell lines used in this study also harbor other oncogenic mutations, such as kRAS mutations in A549 and H23, Keap1 mutations in A549 and H1435, which link to autophagy with p62 and Nrf2 regulation, and pTEN mutation in H23. The authors should discuss this part broadly.

(3) The discussion in paragraph 3 is not precisely. The authors only used 1 uM of B5 compound for the transwell migration assay and EMT. In addition, it is not clear whether the 0.5 uM of B5 compound also inhibited cell migration in the wound healing assay. The experimental setting for migration assay in this study is different to the experiment setting in the study from Pent et al..

6. According to the MDPI guideline on authorship “Those who contributed to the work but do not qualify for authorship should be listed in the acknowledgments.”, the contribution of Hsiang-Hao Chuang in this study should be mentioned in the “Author contributions” section, or his/her name should be moved to acknowledgment.
